# Antibacterial Potential Analysis of Novel α-Helix Peptides in the Chinese Wolf Spider *Lycosa sinensis*

**DOI:** 10.3390/pharmaceutics14112540

**Published:** 2022-11-21

**Authors:** Huaxin Tan, Junyao Wang, Yuxin Song, Sisi Liu, Ziyan Lu, Haodang Luo, Xing Tang

**Affiliations:** 1Department of Biochemistry and Molecular Biology, The Key Laboratory of Ecological Environment and Critical Human Diseases Prevention of Hunan Province Department of Education, School of Basic Medicine, Hengyang Medical School, University of South China, Hengyang 421001, China; 2Hunan Key Laboratory for Conservation and Utilization of Biological Resources in the Nanyue Mountainous Region, College of Life Sciences, Hengyang Normal University, Hengyang 421002, China; 3Department of Clinical Laboratory, The Affiliated Nanhua Hospital, Hengyang Medical School, University of South China, Hengyang 421001, China

**Keywords:** *Lycosa sinensis*, antimicrobial peptide, drug-resistance, biofilm, traditional antibiotic, antibacterial activity

## Abstract

The spider *Lycosa sinensis* represents a burrowing wolf spider (family *Lycosidae*) widely distributed in the cotton region of northern China, whose venom is rich in various bioactive peptides. In previous study, we used a combination strategy of peptidomic and transcriptomic analyses to systematically screen and identify potential antimicrobial peptides (AMPs) in *Lycosa sinensis* venom that matched the α-helix structures. In this work, the three peptides (LS-AMP-E1, LS-AMP-F1, and LS-AMP-G1) were subjected to sequence analysis of the physicochemical properties and helical wheel projection, and then six common clinical pathogenic bacteria (*Enterococcus faecium*, *Staphylococcus aureus*, *Klebsiella pneumoniae*, *Acinetobacter baumannii*, *Pseudomonas aeruginosa*, and *Enterobacter species*) with multiple drug-resistance were isolated and cultured for the evaluation and analysis of antimicrobial activity of these peptides. The results showed that two peptides (LS-AMP-E1 and LS-AMP-F1) had different inhibitory activity against six clinical drug-resistant bacteria; they can effectively inhibit the formation of biofilm and have no obvious hemolytic effect. Moreover, both LS-AMP-E1 and LS-AMP-F1 exhibited varying degrees of synergistic therapeutic effects with traditional antibiotics (azithromycin, erythromycin, and doxycycline), significantly reducing the working concentration of antibiotics and AMPs. In terms of antimicrobial mechanisms, LS-AMP-E1 and LS-AMP-F1 destroyed the integrity of bacterial cell membranes in a short period of time and completely inhibited bacterial growth within 10 min of action. Meanwhile, high concentrations of Mg^2+^ effectively reduced the antibacterial activity of LS-AMP-E1 and LS-AMP-F1. Together, it suggested that the two peptides interact directly on bacterial cell membranes. Taken together, bioinformatic and functional analyses in the present work sheds light on the structure–function relationships of LS-AMPs, and facilitates the discovery and clinical application of novel AMPs.

## 1. Introduction

The frequent and unrestricted use of antibiotics has led to the emergence of bacterial resistance for decades. Recent research by the U.S. Centers for Disease Control and Prevention (CDC) indicates that antibiotic resistance causes millions of infections annually around the world. By 2050, the number of deaths attributable to antibiotic resistance is estimated to reach tens of millions. Consequently, the development of new antibacterial drugs has become a pressing concern [1,2,3].

Small proteins known as antimicrobial peptides (AMPs) have been found to possess antibacterial, antiviral, and antifungal effects [4]. AMPs are ubiquitous in the epithelial barriers of multicellular eukaryotes and forming defenses against external infections. Most of them are short peptides with a net positive charge that attracts them to the membranes of bacteria, which are typically negatively charged [5]. AMPs have demonstrated unparalleled potential as biodrugs against multidrug-resistant bacteria. Unlike traditional antibiotics, which interfere with the metabolic processes of pathogenic microorganisms, AMPs typically exert their antimicrobial effects by physically disrupting microbial cell membrane lipids and inducing leakage of cell contents; thus, they have little impact on resistance evolution in bacteria [6,7,8]. To date, more than 3400 AMPs have been recorded in the Antimicrobial Peptide Database (APD), covering antibacterial, antifungal, antiviral, and antiparasitic functions, and dozens of AMPs are currently being evaluated in clinical trials.

From an evolutionary perspective, AMPs are highly diverse within and across species, with most plant and animal genomes containing 5 to 10 unique AMP gene families ranging in size from 1 to more than 15 paralogous genes [4]. Their diversity is a result of rapid evolution between species, as exemplified by the diversity of AMPs among the speciose groups of toxic animals. Toxin peptide from venomous animals (such as spiders, snakes etc.) is a significant source of naturally active medicinal molecules [9,10]. Spider venom contains a variety of bioactive components, with neurotoxic peptide being the most prominent [11]. On the other side, studies and explorations of spider AMPs are growing [12,13,14]. Most of these peptides are linearly cationic and possess α-helical structures with no more than 80 amino acid residues. The amphiphilic α-helical structure plays a crucial role in the interaction with the cell membrane and further enhances permeability or membrane lysis, which ultimately results in cell death [15]. A 23-amino acid peptide, designated Lycosin-I, with strong antibacterial and antitumor activities was isolated from the venom of *Lycosa singoriensis* (*L. singoriensis*) by Liang’s team [3,16]. Our previous report also showed that Lycosin-I has good potential to be a novel anticancer drug candidate with huge potential in the construction of novel multifunctional antitumor nanomaterials [17]. From the same spider species, Wang et al. isolated a 21-amino acid peptide named Lycosin-II. Although it also belongs to the cationic amphipathic α-helical peptide family, Lycosin-II showed stronger action activity against clinically isolated drug-resistant strains [18]. This suggests that the differences in the sequences of natural AMPs enrich the diversity of their biological activities.

Currently, less than 150 antimicrobial peptides are found in spider venom. Most of these AMPs were isolated from the venom of six *Lycosoidea* spiders [13,14,15,19,20]. The Chinese wolf spider *Lycosa sinensis* (*L. sinensis*) is a burrowing wolf spider widely distributed in the cotton region of northern China, and it belongs to the same genus (*Lycosa*) as the *L. singoriensis* mentioned earlier. In our earlier investigation, the venom components of the two spiders were diverse regardless of their close affinity. Through a combinational strategy on peptidomics and transcriptomics, we identified a total of 52 potential antimicrobial peptide sequences in our previous study. Based on phylogenetic analysis and sequence homology, these potential antimicrobial peptides could be classified into eight different families [21]. However, it is still inconclusive whether they have legitimate antimicrobial activity and how the biological activities and mechanisms diverge between different peptide families.

To better comprehend the sequence–functional diversity relationship of antimicrobial peptides and to identify potential antimicrobial drug molecules with clinical translational value, we conducted a systematic functional validation and comparative analysis of three peptides derived from the LS-AMP family using multiple standard and clinical bacterial strains.

## 2. Materials and Methods

### 2.1. Peptides Synthesis and Characterization

The physiochemical properties of the peptides were checked using ProtParam (https://web.expasy.org/protparam/), and HeliQuest (https://heliquest.ipmc.cnrs.fr/) was used for the helical wheel projection of the peptides. All analysis results were accessed on 21 February 2022.

The peptides were obtained in a powder form by CHENPEPTIDE (Nanjing, China) and synthesized using standard 9-fluoromethoxy carbonyl chemistry in solid phase. The peptides were purified by reversed-phase high performance liquid chromatography (RP-HPLC), and verified by mass spectrometry and high performance liquid chromatography (HPLC) for quality and purity. Purified peptides were dissolved in sterile deionized water and stored at −20 °C.

### 2.2. Materials and Bacterial Strains

*Escherichia coli* (*E. coli*) CCTCC AB 2012883, *E. coli* CCTCC AB 2018675, *Shigella dysenteriae* (*S. dysenteriae*) CGMCC 1.1869, *Proteus vulgaris* (*P. vulgaris*) CGMCC 1.1651, *Salmonella typhimurium* (*S. typhimurium*) CGMCC 1.1174, and *Staphylococcus albus* (*S. albus*) CGMCC 1.3374 were obtained from the China General Microbiological Culture Collection Center. *Staphylococcus aureus* (*S. aureus*) CMCC 26003 was acquired from National Center for Medical Culture Collections. *Staphylococcus citreus* (*S. citreus*) CCTCC AB 91115 was acquired from China Center for Type Culture Collection.

All clinical isolates tested in this study were collected from the Affiliated Nanhua Hospital, University of South China, during the period between September in 2021 and May in 2022. *E. faecium*, *S. aureus*, *K. pneumoniae*, *A. baumannii*, *P. aeruginosa*, and *E. species* clinical multidrug-resistant (MDR) strains isolates were collected from patients’ secretions, urine, sputum, or blood. All bacteria were incubated in Luria-Bertani (LB) medium and Mueller–Hinton broth (MHB) at 37 °C at 200 rpm to logarithmic growth phase and diluted to the desired inoculum concentration, based on the optical density at 600 nm. Azithromycin (AZI), erythromycin (ERY), and doxycycline (DOX) were purchased from APExBIO. All other reagents were obtained from commercial sources and were of analytical grade.

### 2.3. Bacterial Susceptibility Assay

The minimum inhibitory concentration (MIC) values of LS-AMPs by the broth micro-dilution method referring to the Clinical and Laboratory Standards Institute (CLSI) protocol. Bacteria were cultured in LB medium to the exponential phase. MH broth was used to dilute the bacteria to about 10^5^ CFU/mL. Two-fold serial dilutions of LS-AMPs were prepared in 96-well plates. Then, peptide dilutions were mixed with bacterial culture with final peptide concentrations ranging from 0.8 to 50 µM (with standard strains) and 0.8 to 100 µM (with clinical isolates) for 24 h co-incubation at 37 °C, respectively. Positive controls were incubated with Lycosin-I and negative controls were incubated with PBS. Three parallel tests were performed for each microorganism. OD_600_ was measured using a microplate reader and the percentage inhibition was calculated by the following formula:% inhibition = (A_control_ − A_sample_)/A_control_

### 2.4. Inhibit the Formation of Biofilm

The effects of the selected peptide on *E. coli* biofilm formation were assessed using the modified microdilution method [22,23]. *E. coli* CCTCC AB 2012883 cultured overnight at 37 °C were diluted to 10^5^ CFU/mL with fresh LB medium after cells had grown to logarithmic growth phase. A total of 100 µL of bacterial solution was added to a 96-well plate containing 100 µL of continuously 2-fold dilution of various antibacterial substances and cultured at 37 °C for 24 h. The pore solution was slowly sucked out and washed twice by PBS to remove the unattached cells. After adding 200 µL/well methanol to fix 15 min, the methanol solution was sucked out and the drying plate was placed statically. After dyeing with 200 µL 0.1% (*g*/*v*) crystal violet for 10 min, the excess crystal violet was gently washed off with water and the plate was dried. Evaporating crystal violet stains were dissolved by 200 µL 33% (*v*/*v*) acetic acid, and the absorbance was then read at 595 nm. The final values were obtained from the data of three experiments.

### 2.5. Hemolytic Assay

The hemolytic activity of the peptide on mouse erythrocytes was measured by measuring the amount of hemoglobin released. A total of 1% fresh mouse erythrocytes were washed 3 times in PBS (1500 rpm, 10 min), and resuspended with PBS. Different concentrations of peptides (PBS diluted, final concentration 1.6–200 µM) were then mixed with the same volume of mouse erythrocytes suspension, while PBS and double-distilled water were added as negative and positive controls, respectively. After incubation at 37 °C for 1 h, the centrifuged (2000 rpm, 5 min) supernatant was transferred to a new 96-well plate well, and hemoglobin release was monitored by measuring absorbance at 570 nm using a microplate reader. The experiment was performed three independent times. The percentage of hemolysis was calculated as the following equation:% hemolysis = (A_sample_ − A_negative_)/(A_positive_ − A_negative_) × 100%

### 2.6. Calculation and Evaluation of FIC Index

The activities of the AMPs in combination with the antibiotics were analyzed using the checkerboard broth dilution method to determine the fractional inhibitory concentration indices (FICIs). *E. coli* CCTCC AB 2018675, *S. albus* CGMCC 1.3374, *A. baumannii* (1055), and *E. faecium* (1320) were used as the tested strains. In brief, 2-fold serial dilutions of the antibiotic and peptides were prepared at various concentrations into the wells of the sterile 96-well microtiter plate. Additionally, plates were inoculated with bacterial suspension and cultured at 37 °C for 16–24 h. Furthermore, the FIC index value was calculated according to the following equation: FICI = MIC of drug A in combination/MIC of drug A alone + MIC of drug B in combination/MIC of drug B alone. If the MIC was not measurable in this study, the maximum concentration was doubled for calculation. The calculated FICI was interpreted as synergistic (FICI ≤ 0.5), additive (0.5 < FICI < 1), indifferent (1 ≤ FICI < 4.0), or antagonistic (FICI ≥ 4.0).

### 2.7. Scanning Electron Microscopy

*S. albus* CGMCC 1.3374 was grown to the exponential-phase in LB liquid medium. LS-AMP-F1(5× MIC final concentration) was incubated with bacterial suspension with continuous shaking at 37 °C for 10 and 30 min and then centrifuged at 4000 rpm for 10 min. The pellet was then washed with 0.1 M PBS 3 times and fixed with 4% glutaraldehyde at 4 °C overnight. The bacteria were dehydrated by using a grade series of ethanol ranging from 20 to 100% and centrifuged at 10,000 rpm for 10 min. The samples were then suspended in 100% ethanol. The samples were mounted onto aluminum stubs. After sputter-coating with gold, they were analyzed by the scanning electron microscope.

### 2.8. Salt Tolerance

To determine the effect of Mg^2+^ on the antibacterial activity of LS-AMPs, the minimal inhibitory concentrations of LS-AMPs against an *E. coli* CCTCC AB 2012883 strain were determined with or without 5 µM of MgCl_2_, respectively.

### 2.9. Time–Kill Kinetics

The time–kill kinetics of the peptide was studied against a representative strain, *E. coli* CCTCC AB 2012883. The overnight grown inoculum was added to fresh medium and grown to 10^8^ CFU/mL suspensions. The microbial suspension was then incubated with the peptide at 2× and 5× MIC. At 0, 2, 3, 5, 10, 20, 30, and 60 min, the bacterial suspension was serially diluted and spread on LB agar. After the incubation at 37 °C for 24 h, colonies were counted and indicated as log_10_ CFU/mL of each experiment that was repeated thrice.

## 3. Results

### 3.1. Analyses of Peptides Sequence and Physicochemical Property

In our early works, 52 cationic mature peptide sequences without disulfide bonds were identified in *L. sinensis*, which were grouped into eight different families based on phylogenetic analysis and sequence homology. Nonetheless, the biological function of these peptides remains unknown. In this work, three peptides were chosen for synthesis on the basis of their fewer sequence similarity (lower than 50%) to other characterized antibacterial peptides (Appendix A). The sequence information and calculated physicochemical properties of peptides were showed in Figure 1A, and high-performance liquid chromatography (HPLC) and mass spectrometry (MS) profiles are shown in Appendix A. The well-studied antimicrobial and anticancer peptide Lycosin-I, from another wolf spider *L. singorensis*, was selected as positive control for comprehensive assessment of their antibacterial potential. All three peptides contained 18–19 amino acid residues and more than 3 positive charges, exhibiting excellent water solubility (GRAVY: grand average of hydropathicity, ranging from about −0.4 to + 2.0) and thermostability (higher aliphatic index). Peptide-membrane interactions are strongly affected by the amphiphilic structure of peptides. The amphiphilic property of peptides enables amino acids to create hydrophobic moments on the sides of the helix. Hence, the hydrophobic moments (*µH*) of peptides can be utilized to represent their amphiphilic nature [24]. The *µH* and helical wheels of the peptides were predicted using HeliQuest and shown in Figure 1B–E. Similar to Lycosin-I, all three LS-AMP peptides’ hydrophobic and hydrophilic amino acids are distributed in the hydrophobic and hydrophilic sections, respectively, resulting in an amphiphilic α-helical structure for the entire molecule. Even though the hydrophobic surfaces of all three LS-AMP were orientated in the same direction, the hydrophobic amino acids in LS-AMP-G1 were more separated than those in LS-AMP-E1 and LS-AMP-F1.

### 3.2. The Antimicrobial Activities of LS-AMPs

A total of 8 standard reference strains, including 3 Gram-positive and 5 Gram-negative bacteria, were chosen for an initial evaluation on the antimicrobial activity of selected peptides. As indicated in Figure 2A, LS-AMP-E1 and LS-AMP-F1 exhibited similar antibacterial effects towards tested microbes. The MIC values of LS-AMP-E1 and LS-AMP-F1 against 6 tested strands ranged between 50 and 6.25 µM, except for *P. vulgaris* and *S. aureus* (not detected within 50 µM). LS-AMP-G1, on the other hand, had no discernible effect on the growth of any one of the 8 reference strains.

To further evaluate the potential for therapeutic application, we tested the activity of LS-AMPs against an ESKAPE (*E. faecium*, *S. aureus*, *K. pneumoniae*, *A. baumannii*, *P. aeruginosa*, and *E. species*) panel of multidrug-resistant (MDR) pathogens, which are frequently responsible for nosocomial infections. The clinical profiles of MDR strains isolated from 30 cases are listed in Appendix A. As the MIC data shown in Figure 2B, LS-AMP-F1 possess the most potent and extensive bactericidal action among 3 LS-AMPs. Among all 30 tested clinic pathogens, LS-AMP-F1 was the most effective against *A. baumannii,* with the minimum MIC of 3.1 µM (*A. baumannii* 1025 and 1038), which is roughly comparable to that of the positive control Lycosin-I. Yet, the antimicrobial action of LS-AMP-F1 varied significantly amongst bacterial strains. A total of 5 of the 30 clinical isolates tested were insensitive to LS-AMP-F1, while the MICs of LS-AMP-F1 were also highly variable to different isolated strains of the same MDR pathogens. In comparison with LS-AMP-F1, the LS-AMP-E1 was less active, since nearly half of tested microbes were insusceptible to it and the rest measured MICs were mostly greater than or equal to 50 µM. Still, the LS-AMP-G1 remained inactive towards all the tested strains. The cytotoxicity of LS-AMP-E1 and LS-AMP-F1 towards mammalian cells was evaluated by analyzing their hemolytic activity in mouse erythrocytes at various doses (Appendix A). Though, both tested peptides showed dose-dependent hemolytic effects, LS-AMP-F1 have more negligible hemolytic activity with 17% hemolysis detected at 200 µM, indicating that the susceptibility of erythrocytes to LS-AMP-F1 was lower than that of microbial cells.

### 3.3. The Anti-Biofilm Effects of LS-AMPs

Biofilms are an often-ignored cause of chronic and recurrent infections in antibiotic research [25]. They are sessile communities of attached bacterial cells embedded in extracellular polymeric substances (EPS) contributing to disease pathogenicity and drug resistance development [26]. *E. coli* is an important pathogen of hospital-acquired infections and a common cause of biofilm infections. Biofilm-forming *E. coli* are highly drug-resistant and can escape the immune system rendering them susceptible to chronicity and difficult to control. Given that, *E. coli* cells were treated with various concentrations of LS-AMPs, and the biofilm biomass was quantified using a crystal violet staining assay. As revealed in Figure 3, LS-AMP-E1, LS-AMP-F1, and Lycosin-I shared the similar concentration-dependent biofilm inhibitory activity towards *E. coli* cells. At the MIC levels, LS-AMP-E1 and LS-AMP-F1 impeded biofilm formation by 74.7% (12.5 µM) and 73.8% (25 µM) respectively, while Lycosin-I suppressed biofilm formation by 68.2% (3.1 µM). It is noteworthy that LS-AMP-E1 and LS-AMP-F1 were still effective in preventing biofilm formation albeit at sub-MIC concentrations. For example, at the concentration of 3.1 µM, the inhibition rate of LS-AMP-E1 and LS-AMP-F1 on biofilm formation remained at 43.7% and 35.3%, respectively. However, at this concentration, the two peptides did not impair planktonic *E. coli* cell growth, suggesting the distinction between the biofilm-inhibiting and antimicrobial effects of LS-AMP-E1 and LS-AMP-F1, especially at lower peptide concentrations. In context with MIC, the LS-AMP-G1 was inactive against neither attached nor planktonic *E. coli* cells. For Lycosin-I, its biofilm inhibitory effect underwent a significant phase increase with increasing concentration, which may be due to its strong membrane-rupturing effect at higher concentrations.

### 3.4. The Synergistic Effects between LS-AMPs and Antibiotics

Given the anti-biofilm formation potential of LS-AMP-E1 and LS-AMP-F1 manifested at low concentrations, we wonder if the combined utilization of peptides with antibiotics at sub-MIC concentrations could effectively combat the resistance of MDR bacteria. Therefore, we employed the microdilution checkerboard method to assess the synergistic therapeutic effects of peptides with three common antibiotics. Briefly, the MICs of peptides and antibiotics were remeasured when they were used in combination at a series of fixed sub-MIC concentrations (peptides or antibiotics alone). The MICs in combination were recorded in Appendix A, and the calculated FICIs were summarized in Table 1. For the selected two standard strains (*E. coli* and *S. albus*), combining LS-AMP-F1 with any of the three antibiotics (AZI, ERY and DOX) all exerted an effective synergistic effect. In the majority of cases examined, the LS-AMP-E1 exhibited antibiotic synergy comparable to that of the LS-AMP-F1. In certain instances, however, the synergistic effect of LS-AMP-E1 with antibiotics was compromised, as indicated by the FICI values greater than 0.5 (LS-AMP-E1 plus AZI against *E. coli*) and 1.0 (LS-AMP-E1 plus DOX against *S. albus*). With the exception of Lycosin-I-DOX, most combinations of Lycosin-I and tested antibiotics had additive effects on *E. coli*. Intriguingly, the susceptibility of different bacteria to the same antibiotic-peptide combination varies. When DOX was combined with Lycosin-I or LS-AMP-E1, *E. coli* was killed synergistically, but the facilitative relationship between the peptide and the antibiotic was indifferent for *S. albus*. This suggested that the mechanisms underlying bacterial drug tolerance were complex and variable.

To investigate the potential of LS-AMPs to combat drug resistance in clinical situations, we selected clinical isolated MDR *A. baumannii* (1055) as the test strain and calculated FICI values for different combinations. This clinical isolate exhibited significant resistance to DOX (MIC alone ≥16 µg/mL), and the inhibition rate data of the checkerboard test were presented in Figure 4. Both LS-AMP-E1 and LS-AMP-F1 substantially increased the susceptibility of the bacteria to DOX at peptide concentrations below the MIC. In terms of inhibition rate alone, LS-AMP-E1 and LS-AMP-F1 at concentrations of 3.1 µM (1/16× and 1/4× MIC with peptide alone) inhibited 93.8 and 84.7% growth of bacteria, respectively, and elevated the sensitivity of bacteria to DOX by 4 times. In addition, we examined the effect of LS-AMP-F1 on ERY resistance in clinical MDR *E. faecium* (Appendix A). At a concentration of 3.1 µM (1/4× MIC with peptide alone), LS-AMP-F1 was able to completely counteract the bacterial resistance to ERY. Even when using the lowest concentration of ERY (31.25 ng/mL, 1/256 MIC), 89.8% of bacterial growth was still completely inhibited. These data clearly demonstrated the synergistic effects of LS-AMPs with antibiotics, which can effectively improve the resistance of clinical bacteria and has potential application for the treatment of MDR bacteria.

### 3.5. The Rapid Antimicrobial Activities and Mechanisms of LS-AMPs

To evaluate the potential mechanism of LS-AMP-F1 acting on bacteria, scanning electron microscopy (SEM) was used to study membrane morphology. The *S. albus* CGMCC 1.3374 was selected as tested cells to better visualize morphological changes on cell membranes, since the MIC of LS-AMP-F1 against standard *S. albus* was the lowest as shown in Figure 2A (6.25 µM). As depicted in Figure 5A–C, in comparison to the smooth cell surface of the untreated control, *S. albus* cells treated with LS-AMP-F1 for 10 min exhibited a rough appearance and formed pores and fissures on their surface. The collapse of numerous cells caused the release of cellular contents and cellular adhesion Figure 5B. A high proportion of bacterial cell membranes shrank, collapsed, and ruptured after 30 min of treatment with the peptide. The adhesion condition deteriorated further, and treated cells lost entirely normal cellular structure.

Rapid cytotoxicity was also observed with LS-AMP-F1 in Gram-negative *E. coli*. As illustrated in the time–killing curves, bacteria died immediately after being exposed to LS-AMP-F1 at a 5-fold MIC Figure 5D. During the first 5 min of peptide treatment, the number of visible colonies reduced substantially. In addition, after a 10 min incubation with LS-AMP-F1, a gradual reduction of visible colonies was observed. Our findings were consistent with past reports of the fast antibacterial action of AMPs. The rapid killing impact of AMPs on microbes was most likely responsible for the difficulty in establishing resistance to them [4,27,28].

To verify the affinity of LS-AMP-F1 for the bacterial cell membrane, the MIC of LS-AMP-E1 and LS-AMP-F1 against *E. coli* was determined in the presence or absence of 5 µM Mg^2+^. As shown in Appendix A, the antibacterial activity of tested peptides were totally neutralized by the addition of Mg^2+^, and cell proliferation remained unaffected at peptide concentration of 50 µM. This implied that Mg^2+^ may compete with other cations for binding sites on the surface of Gram-negative bacteria, whose outer envelopes are composed of lipopolysaccharides (LPS), which provide more binding sites for divalent cations.

## 4. Discussion

Chronic and recurrent infections caused by multidrug-resistant pathogenic bacteria pose a significant threat to public health due to the rising abuse of antibiotics around the world. Consequently, it has become essential to discover and exploit promising therapeutic agents with novel antibacterial mechanisms [29,30]. In our earlier research, we screened a number of potential antimicrobial peptide sequences by peptidomic and transcriptomic approaches, relying on the high-resolution separation techniques and sequencing analysis. These peptides lack cysteine in their sequences and all contain 15–30 amino acid residues matching with the sequence specificity of linear antimicrobial peptide in APD, although their genuine antibacterial properties lack practical research and analysis [21]. In this study, we selected three LS-AMPs based on their charge, hydrophobicity, and helicity and compared their antimicrobial efficacy, notably for multidrug-resistant clinical strains. Two of the three peptides displayed broad-spectrum bactericidal activity and inhibited the growth of various bacteria at 3.1–50 µM. At lower concentrations, LS-AMP-E1 and LS-AMP-F1 could not only successfully prevent the development of bacterial biofilms, but also exert synergistic therapeutic effects with a variety of antibiotics and even completely counteract the resistance of MDR bacteria, resulting in an improved therapeutic potential. At high concentrations, the two LS-AMPs displayed rapid bactericidal activity and disrupted bacterial cell membranes within 10 min, killing the microorganisms and making it difficult for them to evolve resistance. The findings of this study suggest that LS-AMP-E1 and LS-AMP-F1 possessed great potential to be developed into novel antimicrobial drug candidates by means of further modification.

By comparing the variations in the basic sequences and biological roles of LS-AMPs and Lycosin-I, our work contributes to understanding the structure-function relationships of antimicrobial peptides. The four peptides selected in this work, which all have positively charged and linear sequence characteristics, are capable of forming amphipathic helical structures. The discrepancies in their physicochemical qualities, however, result in their distinct biological activities. Several structural parameters, including net charges, hydrophobicity, amphiphilicity, and structural propensity, affect the antimicrobial activity of AMPs [31,32,33].

Specifically, based on the data of this work, we hypothesized that hydrophobicity, helical structure, and cationicity are critical for the actual antimicrobial activity of the three LS-AMPs. Hydrophobicity, one of the important parameters determining the activity of AMPs, is usually defined as the percentage of hydrophobic residues in the peptide, which determines the distribution of AMPs in the hydrophobic nucleus of the membrane [34,35]. By calculating the hydrophobicity value of each amino acid residue, LS-AMP-F1 and LS-AMP-E1 have relatively high GRAVY values, while LS-AMP-G1 has the lowest GRAVY value, which indicates that LS-AMP-E1 and LS-AMP-F1 should have stronger cell membrane action. In fact, our results are consistent with it. On the other hand, the amphiphilic helix structure is also crucial for the activity of AMPs in the a-helix. Amphiphilicity refers to the degree of spatial separation of hydrophobic and hydrophilic residues on the other side of the molecular framework, which can be quantified by hydrophobic moments [24,36]. Similarly, the *µH* values of LS-AMP-E1 and LS-AMP-F1 are also relatively high, again in accordance with our experimental validation results. This secondary structure-function connection is even more obvious when analyzed in terms of helical wheels. The relatively more active LS-AMP-F1 has a more regular amphipathic helix structure, with its positively charged lysine (K) and hydrophobic amino acid residues distributed on both sides of the helix, respectively. Such a structure is critical for the peptide to interact with bacterial cell membrane. Its positively charged amino acid groups are mainly responsible for the selective recognition binding action with the negatively charged groups on the surface of the bacterial cell membrane, while its hydrophobic surface is crucial for the further interaction of the peptide with the cell membrane [37,38]. Because of this, the absence of LS-AMP-G1 activity is most likely due to its inability to form an effective and stable α-helical structure. The predicted results of its helical wheel suggest that its possible formation of α-helical structure with both hydrophobic and hydrophilic faces is not continuous, both being interrupted by multiple amino acid residues of opposite nature. Additionally, LS-AMP-G1 carries proline (P) residues that are detrimental to the formation of a stable α-helix. In contrast, for the positive control Lycosin-I, its hydrophobic and hydrophobic moments are somewhat attenuated by its higher number of polar amino acid residues (K, glutamic acid E and histidine H). But from the helix wheel, its hydrophilic surface is lined with a large number of positively charged groups, making it easier for its quantity to reach the concentration threshold for membrane rupture when in contact with bacterial cell membranes, thus enhancing the anti-microbial activity. This explains why Lycosin-I has the lowest MIC value against various microorganism. However, there is no linear relationship between one single physicochemical property of peptide and its antimicrobial activity, and antimicrobial activity and selectivity result from a delicate balance between these factors [3].

On the other hand, even for the same peptide, different phenotypic strains exhibit different susceptibilities. One of the necessary conditions for the antimicrobial activity of AMPs is the interaction with negatively charged components of microbial membranes, such as LPS, lipophosphatidic acid (LTA), mannoprotein and phosphatidylinositol in bacterial membranes. In contrast, the evolution of drug resistance is accompanied by changes in the type and content of the various components that make up biological membranes, which greatly complicates the mode and mechanism of action of AMPs [39,40]. Therefore, predicting the activity of AMP based merely on the arrangement of amino acid residues is unreliable. In order to deal with the complicated and severe infection situations, it is vital to integrate the primary structure of AMP with its actual verified function and to continuously optimize the sequence or structure from the natural template for medical applications. 

Finally, from a genetic evolutionary point of view, the conventional thinking has held that AMPs are typically nonspecific, functionally redundant, and largely interchangeable as long as they were produced quickly enough to a level that could restrict infection [4]. In our previous transcriptome analysis of *L. sinensis*, we also found a large number of LS-AMPs with high sequence homology to other AMPs from different species, and multiple mature peptides were separated by regular cleaving signals in the same complex precursor [21]. For example, LS-AMP-E1 and LS-AMP-G1, selected in this paper, are located in a complex precursor. LS-AMP-F1 and other peptides (LS-AMP-B and LS-AMP-C families) are tandemly distributed in another complex precursor, where the mature peptide sequences of the LS-AMP-B and LS-AMP-C families have respectively high sequence similarity with those of Lycosin-I and Lycosin-II from *L. singoriensis*, differing by only one amino acid residue. This phenomenon of rapidly duplicated and pseudogenized antimicrobial peptide genes within and between species has been reported in various studies [41,42,43,44]. It was believed that these gene expression products possess the same function of directly disrupting the microbial cell membrane at a concentration threshold [4]. Contrary to our prior beliefs, the results of this study indicate that genetic variation in AMPs can significantly influence infection resistance. This is evident from the fact that the susceptibility of various bacteria to the four peptides varies greatly. At lower concentrations, some of the tested peptides can influence the formation of bacterial biofilms via a mechanism different from direct membrane disruption. In addition, there are the diverse synergistic effects between these peptides and different antibiotics at even lower concentrations. This all suggests that AMPs are highly functionally diversified and they play roles in varied biological processes, including the regulation of symbiotic communities. Therefore, a comprehensive evaluation and mechanistic investigation of the biological roles of individual peptides is necessary for the practical development of novel antimicrobial peptides, and further research is needed in this field.

## Figures and Tables

**Figure 1 pharmaceutics-14-02540-f001:**
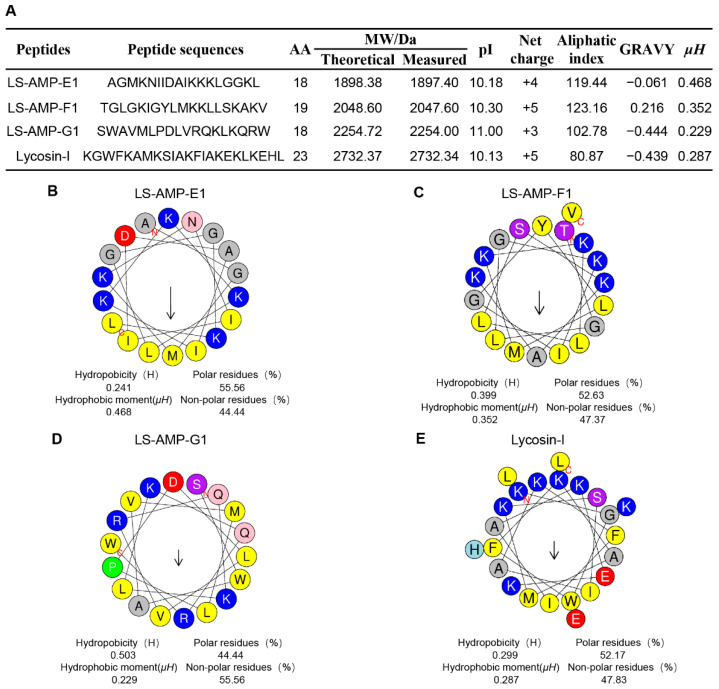
Physicochemical properties (**A**) and helical wheel projection of LS-AMPs (**B**), LS-AMP-E1; (**C**), LS-AMP-F1; (**D**), LS-AMP-G1; (**E**), Lycosin-I. The hydrophobic residues are yellow, positively charged hydrophilic residues are blue, and negatively changed hydrophilic residues are red. The noncharged polar residues are purple. Abbreviations, MW/Da: average molecular mass; pI: isoelectric point; *µH*: hydrophobic moments; GRAVY: grand average of hydropathicity.

**Figure 2 pharmaceutics-14-02540-f002:**
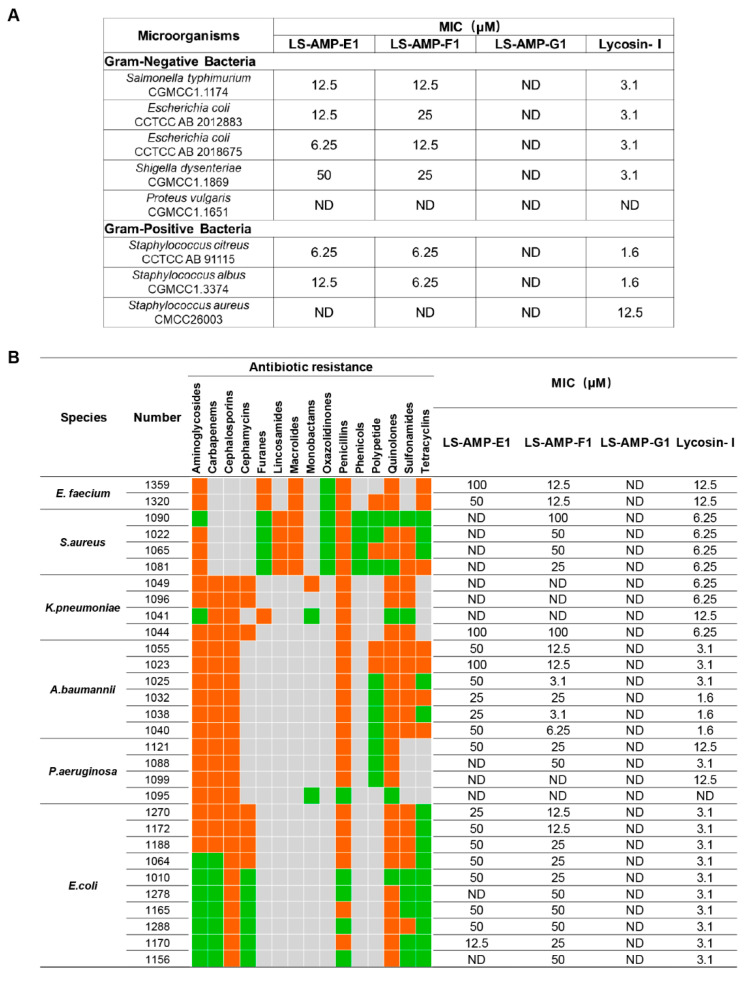
The comparison of antimicrobial activity of LS-AMPs. The MIC values of peptides against standard strains (**A**) and clinical bacterial strains (**B**). Bacteria were susceptible to all (green boxes) or intermediate/resistant to at least one (orange boxes) of the antibiotics per class. Gray boxes are shown if the susceptibility to agents in that class is not assessed. MIC was defined as the minimum concentration to kill the bacterial strains completely. ND: no antimicrobial activity at the concentrations tested.

**Figure 3 pharmaceutics-14-02540-f003:**
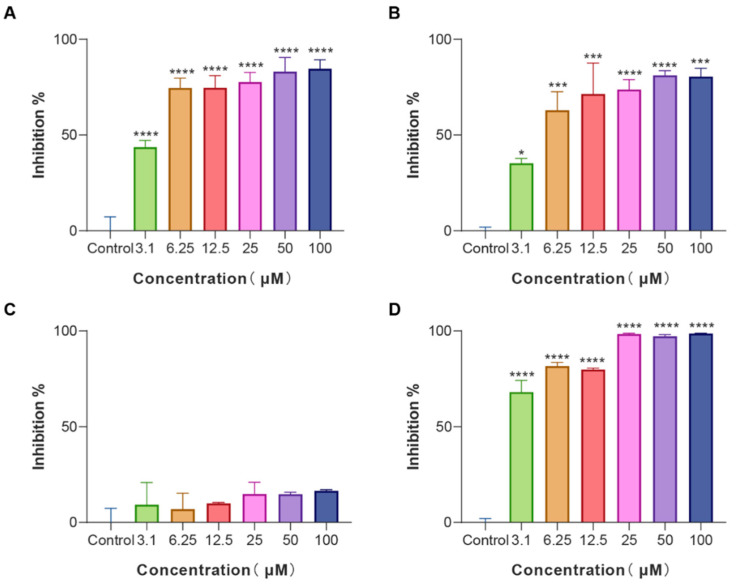
Biofilm inhibition activity of LS-AMPs on *E. coli* biofilm formation (**A**), LS-AMP-E1; (**B**), LS-AMP-F1; (**C**), LS-AMP-G1; (**D**), Lycosin-I. Data represent means ± SE of 3 individual experiments. * *p* < 0.05, *** *p* < 0.001 and **** *p* < 0.0001.

**Figure 4 pharmaceutics-14-02540-f004:**
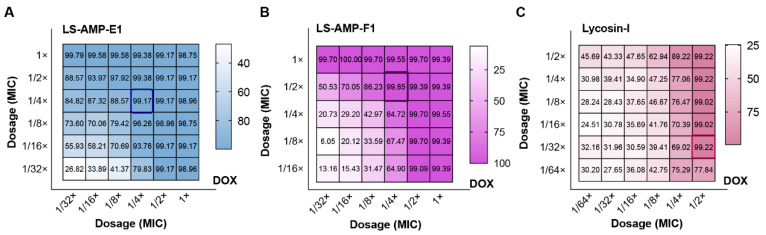
Inhibition rate of the checkerboard test for MDR *A. baumannii* (1055). (**A**), the combination of LS-AMP-E1 and DOX; (**B**), the combination of LS-AMP-F1 and DOX; (**C**), the combination of Lycosin-I and DOX. The dosage was indicated by the dilution multiple of MIC (antibiotic or peptides alone).

**Figure 5 pharmaceutics-14-02540-f005:**
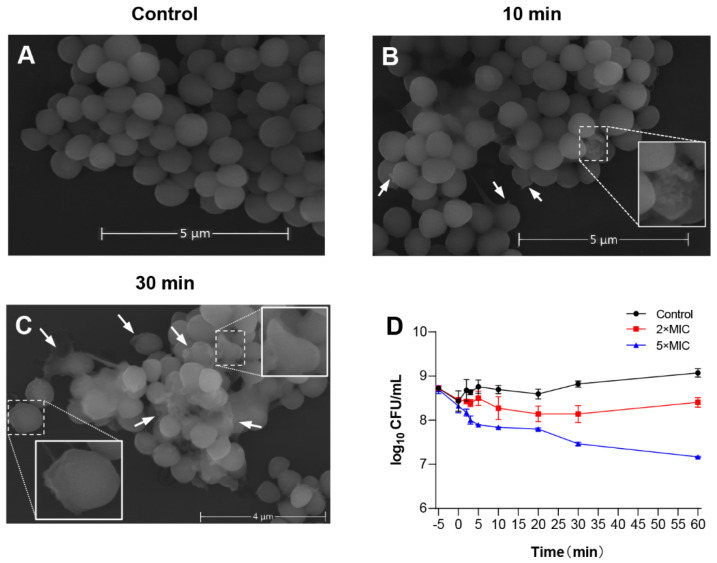
The membrane-permeabilizing effect of LS-AMP-F1. Scanning electron microscopy images of LS-AMP-F1-treated *S. albus*: Negative control (**A**); 10 min after LS-AMP-F1 treatment (**B**); 30 min after LS-AMP-F1 treatment (**C**). The white arrows indicated the cell membrane damages, and the degenerated cells were enlarged in (**B**,(**C**)). Killing kinetics of LS-AMP-F1 against *E. coli* (**D**). The −5 min point represents untreated bacteria, and 0 min represents the time of the first sample collection immediately after the addition of the peptides to the bacterial suspension. The other samples were collected at 5, 10, 20, 30, and 60 min. Bacterial counts represent the average of three dishes.

**Table 1 pharmaceutics-14-02540-t001:** FIC index of the standard strains.

Microorganisms	Antibiotics	FICI		
		LS-AMP-E1	LS-AMP-F1	Lycosin-I
** *E. coli* ** **CCTCC AB 2018675**	Azithromycin	0.56	0.28	0.75
	Erythromycin	0.50	0.27	0.53
	Doxycycline	0.28	0.38	0.27
** *S. albus* ** **CGMCC1.3374**	Azithromycin	0.28	0.50	0.28
	Erythromycin	0.28	0.28	0.28
	Doxycycline	1.03	0.50	1.02

## Data Availability

The data presented in this study are available on request from the corresponding author.

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
