# Peer review of "Antibacterial Potential Analysis of Novel α-Helix Peptides in the Chinese Wolf Spider Lycosa sinensis"

_pharmaceutics, 2022, doi:10.3390/pharmaceutics14112540_

Round 1
Reviewer 1 Report
1. General comments
In the manuscript, antimicrobial activity of three peptides (LS-AMP-E1, LS-AMP-F1, and LS-AMP-G1) against 6 common clinical pathogenic bacteria was investigated, and it was demonstrated that LS-AMP-E1 and LS-AMP-F1 had different inhibitory activity against 6 clinical drug-resistant bacteria with multiple drug-resistance. In addition, LS-AMP-E1 and LS-AMP-F1 exhibited varying degrees of synergistic therapeutic effects with traditional antibiotics, significantly reducing the working concentration of antibiotics and AMPs. The result suggests that LS-AMPs have a potential to be developed into novel antimicrobial drug candidates.
2.Major revision
1) Figure 4 and page 9
It is essential to explain the relationship between “Dosage (MIC) shown as x1, x1/2 and so on in Figure 4” and “a word “3.1 mM” in line 5~8 from the bottom in page 9”.
2) Figure 5
a) It is difficult to understand the degenerated (shrank, collapsed, and ruptured) cells from Figure 5A-C. It is strongly recommended to add the photograph of cells more enlarged.
b) It is essential to exchange “Figure 5E” to “Figure 5D” for explaining the sentences of line 17~24 from the bottom in page 10.
c) It is essential to revise a word “Figure 5D” to “Figure 5E” at line 7 from the bottom in page 10.
d) It is essential to explain the relationship between “0 mM Mg2+” in Figure 5D and the value “50 µM” at the sentence of line 6 from the bottom in page 10. Is 50 µM Mg2+ included in the reaction solution used in Figure 5D?
3. Minor revision
1) Legend of Figure 1A: It is recommended to explain the value of MW/Da shown in Figure 1A. Is it monoisotopic or average molecular mass?
2) Page 9: A word “doxycycline” was used in line 10~16 from the front, but DOX was used in line 7~11 from the bottom. It is recommended to unify doxycycline or DOX.
3) Page 12, line 2~3: Revise “LC-AMPs” to “LS-AMPs”.
Author Response
Response to Reviewer 1 Comments
1.General comments
In the manuscript, antimicrobial activity of three peptides (LS-AMP-E1, LS-AMP-F1, and LS-AMP-G1) against 6 common clinical pathogenic bacteria was investigated, and it was demonstrated that LS-AMP-E1 and LS-AMP-F1 had different inhibitory activity against 6 clinical drug-resistant bacteria with multiple drug-resistance. In addition, LS-AMP-E1 and LS-AMP-F1 exhibited varying degrees of synergistic therapeutic effects with traditional antibiotics, significantly reducing the working concentration of antibiotics and AMPs. The result suggests that LS-AMPs have a potential to be developed into novel antimicrobial drug candidates.
2.Major revision
1) Figure 4 and page 9
It is essential to explain the relationship between “Dosage (MIC) shown as x1, x1/2 and so on in Figure 4” and “a word “3.1 mM” in line 5~8 from the bottom in page 9”.
Response 2.1: Thank you for your valuable suggestions. In the revised version, the explanation of the relationship between the dilution of MIC and the actual concentration is added in the main text.
2) Figure 5
- a) It is difficult to understand the degenerated (shrank, collapsed, and ruptured) cells from Figure 5A-C. It is strongly recommended to add the photograph of cells more enlarged.
- b) It is essential to exchange “Figure 5E” to “Figure 5D” for explaining the sentences of line 17~24 from the bottom in page 10.
- c) It is essential to revise a word “Figure 5D” to “Figure 5E” at line 7 from the bottom in page 10.
- d) It is essential to explain the relationship between “0 mM Mg2+” in Figure 5D and the value “50 µM” at the sentence of line 6 from the bottom in page 10. Is 50 µM Mg2+ included in the reaction solution used in Figure 5D?
Response 2.2: a): Thanks for your suggestions. The enlarged images of degenerated cells have been added in Figure 5 as you suggested.
b-c): Thanks for pointing that out. We apologize for such errors in our first manuscript.
The figure sequence has been adjusted according to the main text, and similar issues have been carefully checked and corrected in the revised version.
d): Thanks for your valuable suggestions. The “50 µM” at line 6 from the bottom in page 10 indicated the peptide concentration. We have revised the phrases in the text as “cell proliferation remained unaffected at peptide concentration of 50 mM” in the revised version.
- Minor revision
1) Legend of Figure 1A: It is recommended to explain the value of MW/Da shown in Figure 1A. Is it monoisotopic or average molecular mass?
Response 3.1: Thanks for your valuable suggestion. The MW/Da value in Figure 1A indicated average molecular mass. The corresponding explanation of the abbreviation has been added to the legend of Figure 1 in revised manuscript.
- Page 9: A word “doxycycline” was used in line 10~16 from the front, but DOX was used in line 7~11 from the bottom. It is recommended to unify doxycycline or DOX.
Response 3.2: Thanks for pointing that out. The correct use of abbreviations has been doubled checked and corrected in our revised manuscript including the correction of “DOX” in page 9 as you suggested.
- Page 12, line 2~3: Revise “LC-AMPs” to “LS-AMPs”.
Response 3.3: Thanks for pointing that out. The spelling and grammatical errors have been carefully checked and corrected in the revised manuscript including the correction of “LS-AMPs” as you suggested.

Reviewer 2 Report
In this manuscript, the authors Tan et al., studied the antimicrobial activity of 3 peptides that were previously isolated from Lycosa sinensis venom. The results obtained were striking. However, substantial modifications must be made to the manuscript before it is published.
Line 19: They mention a deep bioinformatic analysis, not found in the manuscript. Check please.
Line 55: keep reference format
Line 109: Report peptide purity
Line 113-119: Italic letter to name the species of the microorganism
Line 136: clarify why two maximum concentrations
Line 156: Justify why this concentration of erythrocytes. Equivalence of the number of erythrocytes?
Line 159: Please send me a reference where double distilled water has a hemolytic effect. Any solute that modulates osmolarity? please explain
Line 165: should be multiplied x 100.
Line 172: review ".And plates"
Line 180: Contextualize why these strains were used for SEM studies.
Line 206: show algorithm data for similarity analysis
Lines 317-329: paragraph out of the margin with different font size
Line 347: cite past reports
Line 350: The affinity measurement should be considered from a ligand-receptor titration to calculate a Ka. Here, the authors report the effect of the Mg+2 ion on the antimicrobial activity of the peptide, which could not necessarily be due to competition, but also coordination effects with the peptide. Furthermore, I do not find a significant contribution of this section to the interesting results that were obtained previously.
Line 351: italic letter for bacteria
Discussion section: It is a poor discussion, since the results obtained in this study were not discussed. The discussion should address: 1. How the structural variables of the 3 peptides influence each of the studied models. 2. why LS-AMP-G loses its antimicrobial property 3. how different phenotypes of MDR strains influence the MIC value of peptides.
Line 379: The choice of sequences is contradictory, since the results says that they were chosen on the basis of their fewer sequence similarity to other characterized antibacterial peptides, but not because of these structural properties.
No acknowledgments or funding reported. A letter of approval from the ethics committee should be reported, for the authorization of the handling of erythrocytes.
Author Response
Response to Reviewer 2 Comments
In this manuscript, the authors Tan et al., studied the antimicrobial activity of 3 peptides that were previously isolated from Lycosa sinensis venom. The results obtained were striking. However, substantial modifications must be made to the manuscript before it is published.
- Line 19: They mention a deep bioinformatic analysis, not found in the manuscript. Check please.
Response 1: Thanks for pointing that out. The "in-depth bioinformatic analysis" mentioned in the abstract referred to the sequence analysis, physicochemical property calculations, and structure prediction that we described in Figure 1. In addition, we have performed homology comparison of three LS-AMPs sequences through the Antimicrobial Peptide Database (APD) and their homology data with verified antimicrobial peptide sequences are supplemented in the supplementary material. After consideration, we have changed "in-depth bioinformatic analysis" to "sequence analysis", which you pointed out, to make the article more rigorous.
- Line 55: keep reference format
Response 2: The format of the references has been adjusted and standardized in the revised version.
- Line 109: Report peptide purity
Response 3: The purity analysis of the synthesized peptides has been added to Figure S2-4 according to your request. As the profiles of the HPLC shown, the purity of all the peptides synthesized in this study is greater than 95%, which meets the experimental requirements.
- Line 113-119: Italic letter to name the species of the microorganism
Response 4: Thanks for pointing that out. All the name of the species of the microorganism have been double-checked and corrected in Italic including in Line 113-119 as you mentioned.
- Line 136: clarify why two maximum concentrations
Response 5: During the experiments, we found that the susceptibility of clinical multidrug resistant bacteria to peptides was reduced. Therefore, in order to accurately measure the MIC values of peptides for various bacteria, we increased the maximum concentration to 100 mM when testing clinical strains, and the related explanation has been added to the corresponding position in the revised version.
- Line 156: Justify why this concentration of erythrocytes. Equivalence of the number of erythrocytes?
Response 6: The experimental protocols for the hemolysis experiments, including the preparation and dilution of erythrocyte suspensions, were referred to several related articles1 2. The 1% (v/v) erythrocyte concentration was chosen because at this cell concentration, the structure of erythrocytes is more stable and the hemolysis rate is lower, which is suitable for assessing the hemolysis rate of incubated agents. By cell counting, we measured that the 1% erythrocyte concentration was close to 108 cells/mL, which is consistent with the standard criteria for the required number of erythrocytes in the reference literatures3 4. During the experiments, there was indeed no significant occurrence of hemolysis in the 1% erythrocyte negative control, which also proved the feasibility of this erythrocyte concentration for the comparison of hemolytic effect.
- Line 159: Please send me a reference where double distilled water has a hemolytic effect. Any solute that modulates osmolarity? please explain
Response 7: The hemolysis effect is a result of the destruction of the red blood cell membrane, resulting in the efflux of hemoglobin from the red blood cell. Since the ionic concentration (osmotic pressure) of double-distilled water is small or almost non-existent, the difference in ionic concentration between the inside and outside of the cell when acting on red blood cells makes it easy for the cell to absorb water and burst, resulting in the hemolysis reaction. In fact, the use of double-distilled water as a 100% positive control for hemolysis assays has been extensively reported. The references attached to this response are for your reference5 6 7.
In addition to double distilled water, some hypotonic salt solutions such as 0.5% NH4OH, 0.45% NaCl, etc. can be used to regulate cellular osmolarity through a similar principle of water uptake and rupture. Besides, Triton X is also a recognized positive control for hemolysis reaction due to its lysis effect on the cell membrane. In the present work, the reason we chose double distilled water as a positive control was to be consistent with the hemolytic activity assay of Lycosin-I mentioned in the text. In fact, the hemolytic activity of Lycosin-I that we measured in this work is consistent with the data reported previously.
- Line 165: should be multiplied x 100.
Response 8: Thank you for pointing that out. We have made the appropriate changes in the revised version.
- Line 172: review ".And plates"
Response 9: Thank you for pointing that out. We have made the appropriate changes in the revised version.
- Line 180: Contextualize why these strains were used for SEM studies.
Response 10: Based on the MIC data of Figure 2A, we chose S. albus because LS-AMP-F1 had the strongest inhibitory effect with an MIC value of 6.25 mM against it, and peptide may cause more pronounced effects on cell membrane. Together with the fact that the standard strain of S. albus has typical characteristics of Staphylococci, we chose SEM to observe the effects of bacterial cell membrane of S. albus treated with peptides. Based on your suggestion, the related descriptions have been added to the revised version.
- Line 206: show algorithm data for similarity analysis
Response 11: Thank you for your suggestion. In this work, we have used antimicrobial peptide database (ADP) to perform homology matching of the sequences of three LS-AMPs. Based on the database search results, the antimicrobial peptide sequences with the highest homology matched to each peptide sequence were aligned with LS-AMPs. The results of the comparison are added in the modified Figure S1.
- Lines 317-329: paragraph out of the margin with different font size
Response 12: Thanks for pointing that out, the formatting issues have been carefully checked and corrected in the revised manuscript.
- Line 347: cite past reports
Response 13: Thanks for pointing that out. The relevant references have been cited accordingly as you suggested.
- Line 350: The affinity measurement should be considered from a ligand-receptor titration to calculate a Ka. Here, the authors report the effect of the Mg+2 ion on the antimicrobial activity of the peptide, which could not necessarily be due to competition, but also coordination effects with the peptide. Furthermore, I do not find a significant contribution of this section to the interesting results that were obtained previously.
Response 14: Thank you for your valuable suggestions and reasonable analysis. In the present work, we used an additional increase in Mg2+ concentration on the outer side of the bacterial cell membrane to speculate on the potential mechanism of action of LS-AMPs, which was mainly based on the literature of antibacterial studies of Lycosin-I and Lycosin-II1 2. The electrostatic interaction between cationic antimicrobial peptides and negatively charged bacterial cell membranes is the main force for selective binding and adsorption of cationic peptides to bacterial cell membranes. Therefore, increasing the concentration of cations outside the membrane would competitively decrease the binding of peptides to the membrane and further increase the effective concentration of peptides acting on bacteria. However, as you said, the inference from the results in reverse is not well justified and does not correlate well with the previous results. So based on this consideration, we have placed this part of the data in the supplemental material and revised the discussion in the results section more strictly.
- Line 351: italic letter for bacteria
Response 15: Thank you for pointing that out. All the names of microbial species have been carefully checked and corrected in Italic including in Line 351.
- Discussion section: It is a poor discussion, since the results obtained in this study were not discussed. The discussion should address: 1. How the structural variables of the 3 peptides influence each of the studied models. 2. why LS-AMP-G loses its antimicrobial property 3. how different phenotypes of MDR strains influence the MIC value of peptides.
Response 16: Thanks for your valuable suggestion. Based on your suggestion, we have rewritten the Discussion section in the revised version. Regarding your questions, we have made the further discussion in the corresponding place in the revised manuscript.
- Line 379: The choice of sequences is contradictory, since the results says that they were chosen on the basis of their fewer sequence similarity to other characterized antibacterial peptides, but not because of these structural properties.
Response 17: Thank you for your valuable comments. First of all, all potential LS-AMPs we identified previously fit linear cationic peptide sequence characteristics, such as no disulfide bond and rich in multiple basic amino acid residues. In the present work, we performed homology matching of LS-AMPs sequences using the APD to screen peptide sequences with less than 50% sequence similarity to known antimicrobial peptides, which was mainly to screen novel antimicrobial peptide molecules with novel sequence modalities. The information of the related sequence comparison we added in FigureS1.
- No acknowledgments or funding reported. A letter of approval from the ethics committee should be reported, for the authorization of the handling of erythrocytes.
Response 18: Thank you for pointing that out. Additional statements about this work,including Funding information, Institutional Review Board Statement, Conflicts of Interest, have been added in the end of the revised manuscript.
References
- Tan, H.; Ding, X.; Meng, S.; Liu, C.; Wang, H.; Xia, L.; Liu, Z.; Liang, S., Antimicrobial potential of lycosin-I, a cationic and amphiphilic peptide from the venom of the spider Lycosa singorensis. Curr Mol Med 2013, 13 (6), 900-10.
- Wang, Y. J.; Wang, L.; Yang, H. L.; Xiao, H. L.; Farooq, A.; Liu, Z. H.; Hu, M.; Shi, X. L., The Spider Venom Peptide Lycosin-II Has Potent Antimicrobial Activity against Clinically Isolated Bacteria. Toxins 2016, 8 (5).
- Chionis, K.; Krikorian, D.; Koukkou, A. I.; Sakarellos-Daitsiotis, M.; Panou-Pomonis, E., Synthesis and biological activity of lipophilic analogs of the cationic antimicrobial active peptide anoplin. J Pept Sci 2016, 22 (11-12), 731-736.
- Yi, T. H.; Huang, Y. B.; Chen, Y. X., Production of an Antimicrobial Peptide AN5-1 in Escherichia coli and its Dual Mechanisms Against Bacteria. Chem Biol Drug Des 2015, 85 (5), 598-607.
- Hollmanna, A.; Martinez, M.; Noguera, M. E.; Augusto, M. T.; Disalvo, A.; Santos, N. C.; Semorile, L.; Maffia, P. C., Role of amphipathicity and hydrophobicity in the balance between hemolysis and peptide-membrane interactions of three related antimicrobial peptides. Colloid Surface B 2016, 141, 528-536.
- Shi, J.; Liu, Y.; Wang, Y.; Zhang, J.; Zhao, S. F.; Yang, G. L., Biological and immunotoxicity evaluation of antimicrobial peptide-loaded coatings using a layer-by-layer process on titanium. Sci Rep-Uk 2015, 5.
- Maturana, P.; Martinez, M.; Noguera, M. E.; Santos, N. C.; Disalva, E. A.; Semorile, L.; Maffia, P. C.; Hollmann, A., Lipid selectivity in novel antimicrobial peptides: Implication on antimicrobial and hemolytic activity. Colloid Surface B 2017, 153, 152-159.

Round 2
Reviewer 2 Report
The authors have satisfactorily addressed the comments and the manuscript has substantially improved.